# Classification of Transmission Line Corridor Tree Species Based on Drone Data and Machine Learning

Xiuting Li [1,2], Ruirui Wang [1,2,*], Xingwang Chen [1,2], Yiran Li [1,2] and Yunshan Duan [1,2]

1   College of Forestry, Beijing Forestry University, Beijing 100083, China; lixiuting97@bjfu.edu.cn (X.L.);
    15810285986@163.com (X.C.); yiranliyy@163.com (Y.L.); 13383169667@163.com (Y.D.)
2   Beijing Key Laboratory of Precision Forestry, Beijing Forestry University, Beijing 100083, China
*   Correspondence: wangruigis@163.com

**Abstract:** Tree growth in power line corridors poses a threat to power lines and requires regular inspections. In order to achieve sustainable and intelligent management of transmission line corridor forests, a transmission line corridor tree barrier management system is needed, and tree species classification is an important part of this. In order to accurately identify tree species in transmission line corridors, this study combines airborne LiDAR (light detection and ranging) point-cloud data and synchronously acquired high-resolution aerial image data to classify tree species. First, individual-tree segmentation and feature extraction are performed. Then, the random forest (RF) algorithm is used to sort and filter the feature importance. Finally, two non-parametric classification algorithms, RF and support vector machine (SVM), are selected, and 12 classification schemes are designed to perform tree species classification and accuracy evaluation research. The results show that after using RF for feature filtering, the classification results are better than those without feature filtering, and the overall accuracy can be improved by 3.655% on average. The highest classification accuracy is achieved when using SVM after combining a digital orthorectification map (DOM) and LiDAR for feature filtering, with an overall accuracy of 85.16% and a kappa coefficient of 0.79.

**Keywords:** light detection and ranging (LiDAR); individual tree crown delineation; transmission line corridor; random forest (RF); support vector machine (SVM)



## 1. Introduction

Excessive growth of trees around the transmission line corridor tends to obstruct transmission lines. Therefore, trees that grow to a height that threatens transmission lines need to be regularly inspected and removed [1]. In order to achieve sustainable and smart management of forests in transmission line corridors, trees in transmission line corridors are not cut down all at once, but systematically through the establishment of a transmission line corridor tree barrier management system. By inputting tree obstruction information into the information base, a model of tree growth is created to facilitate inquiries about tree obstruction hazards, so that planned felling can be developed. Therefore, it is important to know the tree species. With the continuous development of remote-sensing technology, tree species classification has also been applied to transmission line corridors. However, most of the data sources used in the research on tree species classification of transmission line corridors are single data sources [2], and the classification accuracy is not sufficient to effectively prevent hidden dangers caused by trees in these corridors. The classification of tree species based on multi-source remote sensing has advantages in other fields [3–7], so this study considers using multi-source unoccupied aerial vehicle (UAV) data to classify tree species in transmission line corridors to improve classification accuracy.

Machine learning (ML) algorithms can be used to solve the non-linear sample classification problem of tree species classification. Many scholars have used ML to identify or classify tree species [8–11]. For instance, Franklin et al. [12] used the multi-spectral

data obtained by drones combined with ML algorithms to classify deciduous tree species, with an overall classification accuracy of 78%. Ahmed et al. [13] placed three multispectral cameras on a UAV and used the acquired data to identify Sequoia; the results showed that the identification accuracy was as high as 89%. Chan et al. [14] compared the classification accuracy of different classification algorithms based on hyperspectral data, and the results showed that the classification accuracy of AdaBoost classification and random forest (RF) classification algorithm was almost the same (close to 70%); the difference was less than 1%, which was higher than that of the neural network classifier that has an overall accuracy of 63.7%. Puttonen et al. [15] collected LiDAR data and hyperspectral data at the same time based on the Sensei system of the Finnish Geodetic Institute to classify coniferous and broad-leaved species. The results show that the classification accuracy using only spectral features was 90.5%, while the overall accuracy of classification combined with spectral and structural features reached 95.8%. Considering airborne hyperspectral and LiDAR data obtained at the same time and the support vector machine (SVM) classifier, Liu Yijun et al. [16] classified the dominant tree species in the Pu'er Mountain experimental area forest. The results showed that the overall accuracy of the fusion data classification reached 80.54%, compared with only using spectral information. In summary, the preceding research shows that using multi-source remote-sensing data combined with ML can enable effective identification of tree species. In the past, studies on tree species classification used remote-sensing images with a low-resolution rate, and most of them used a single data source. However, using multiple remote-sensing data sources and ML algorithms to classify tree species represents a research hotspot [2,17–20]. In addition, relatively few studies have been conducted on the classification of tree species in transmission line corridors.

Accurate spatial information on tree species is essential for forestry management and is crucial for sustainable management of forest resources and effective monitoring of species diversity, which can help solve a wide variety of application problems faced by forestry management. In this study, experiments were conducted to address the issue of how to improve the accuracy and efficiency of forest species classification using remote sensing technology. On the one hand, the complementary effect of the superior features of airborne LiDAR point clouds and DOMs (digital orthophoto maps) is realized, and the classification accuracy of woody species is improved by feature screening. In addition, various classification methods are analyzed and compared, which has important theoretical significance. On the other hand, this helps to obtain finer tree species information of the transmission channel more accurately and quickly and provides a reference basis for the tree obstacle potential management system. It is of great practical significance for establishing tree growth models, as well as querying and timely cleaning of tree barrier hazards in transmission line corridors.

This study fully utilizes the advantages of machine classification algorithms in high-dimensional feature classification and solves the problem of low classification accuracy of tree species in transmission line corridors. First, the vertical information provided by the LiDAR data and the horizontal information provided by the DOM are combined to segment the canopy and extract the canopy features. Then, the RF algorithm is used in feature selection. Finally, the RF and SVM algorithms are used to classify tree species, and the high-precision classification of tree species in the transmission line corridor is achieved.

## 2. Materials and Methods

### 2.1. Study Area

The study area is located in the northeastern part of Chizhou city, Anhui Province, with an altitude between 1.8 m and 112.2 m. The geographical position is 117°46'–117°56' east longitude and 30°39'–30°41' north latitude. It has a warm and humid subtropical monsoon climate with four distinct seasons, sufficient rainfall, annual average temperature of 16.5 °C, annual average precipitation of 1400–2200 mm, a long period of sunshine, a short frost-free period, and approximately 40 rainy days. The study area is rich in vegetation types. The dominant tree species include broad-leaved tree species such as fir, bamboo,

maple, and oak, mainly in middle-aged and mature forests. The specific location of the study area is shown in Figure 1.

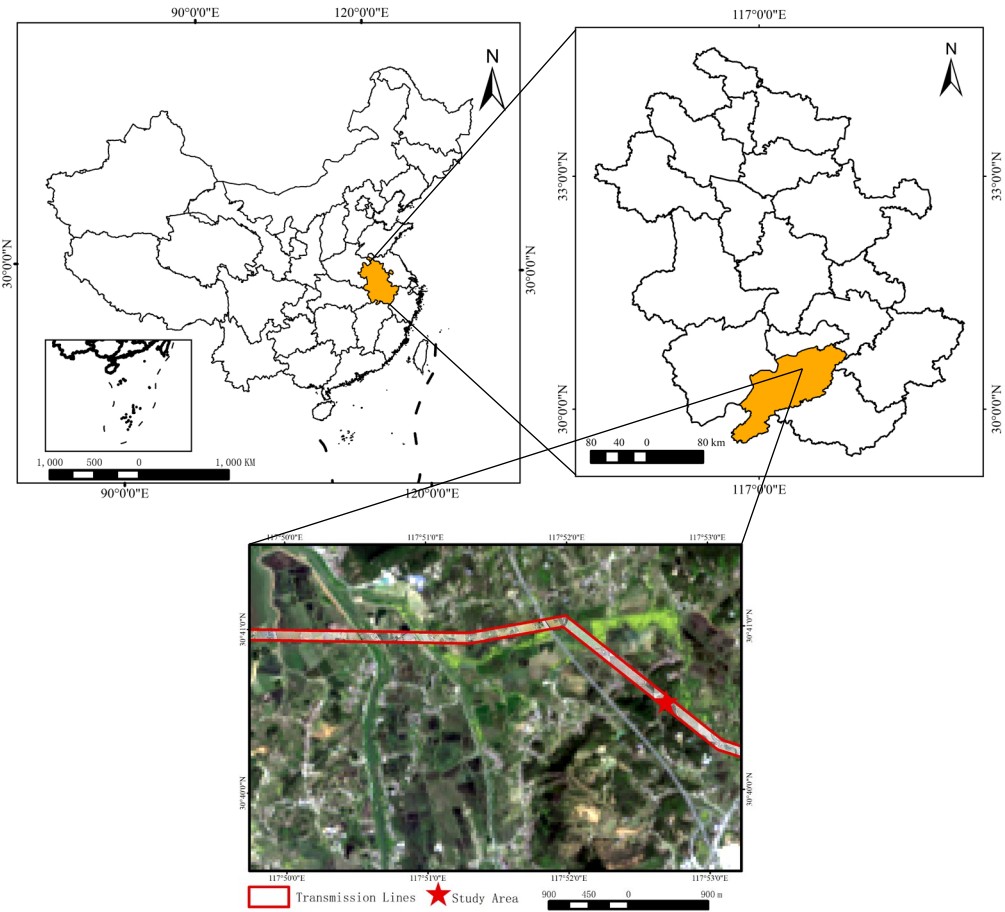

**Figure 1.** Location of the study area.

## 2.2. Aerial Image and LiDAR Data

The data used in this study include airborne LiDAR point-cloud data and synchronized high-resolution digital orthophotos. The flight time was June 2016, under clear weather conditions with good visibility. The airborne LiDAR point-cloud data were collected using the Optech ALTM Galaxy system. The parameters are shown in Table 1. The downlink channel of one of the towers in the study area was selected as the test area. The original LiDAR point-cloud data and orthophotos of the specific study area are shown in Figure 2 and Supplementary Materials File S1.

**Table 1.** The parameters of airborne remote sensing system platform.

| DOM | | LiDAR | |
|---|---|---|---|
| Ground resolution | 0.1 m | Wavelength | 1064 nm |
| Focal length | 35 mm | Laster beam divergence | 0.25 mrad |
| | | Maximum point density | 93 pts/m$^2$ |
| | | Minimum point density | 0.6 pts/m$^2$ |

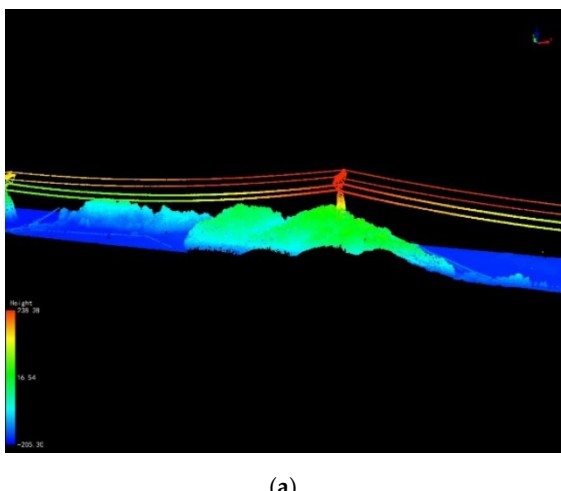

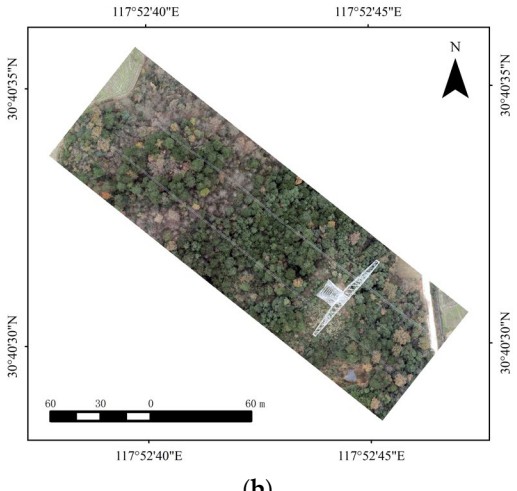

(**a**)                                            (**b**)

**Figure 2.** Data sources: (**a**) point-cloud data graph in the study area and (**b**) digital orthophoto map of the study area.

### 2.3. Methods

This study combines the horizontal characteristics of the DOM and the vertical characteristics of LiDAR data and selects ML algorithms to classify the tree species around the transmission line corridor. The main steps are as follows: (1) LiDAR point-cloud data are used to generate a CHM (canopy height model). (2) The watershed algorithm is used in CHM-based single wood segmentation. (3) The RF algorithm is used to select the best feature combination for individual-tree species classification and analyze and compare the impact of feature se-lection on tree species classification. (4) A classification scheme is designed, the effect of multi-source UAV data in individual-tree species classification is studied, and the ability of different non-parametric learning algorithms is evaluated to classify tree species at the individual tree level. The technical process is shown in Figure 3.

#### 2.3.1. Data Preprocessing

In this study, the LiDAR point cloud data are already classified point clouds. The point clouds of extraneous objects on the ground such as transmission lines and tower bases are removed before the segmentation of individual tree canopies is performed. Only vegetation points and ground points in the point cloud are retained. The ground points in the classified point cloud data are used as feature points to perform interpolation operations to construct a DEM. The first echo points of vegetation points are interpolated, and the difference operation is performed to construct a DSM. The interpolation method uses Triangulation Irregular Network Interpolation (TIN), which constructs triangles from a series of points. The advantage of the TIN method is its ability to preserve surface details in topographically complex areas. The difference operation is performed on the generated DSM and DEM raster data to obtain the canopy height model after elevation normalization. There are black or gray invalid holes in the original CHM caused by abnormal changes in height, which will affect tree vertex detection and tree crown sketching. In this study, the median filter in the smoothing filter is selected for smoothing, a new CHM is generated, and the invalid value of the optimized CHM image is filled.

#### 2.3.2. Individual-Tree Canopy Segmentation

Before individual-tree canopy segmentation, point clouds of irrelevant objects on the ground such as transmission lines and tower bases are removed, and only vegetation points and ground points in the point cloud are retained, thus improving the accuracy of tree segmentation.

Watershed segmentation algorithm is a mathematical morphology segmentation method based on topology theory proposed by Vincent [21]. This algorithm considers im-

age segmentation according to the composition of the watershed and has a good response to weak edges. It is one of the most common segmentation methods. In this paper, the watershed segmentation algorithm is used to segment the single tree canopy for CHM, the Gaussian smoothing factor is 1, and the smoothing window used is $5 \times 5$.

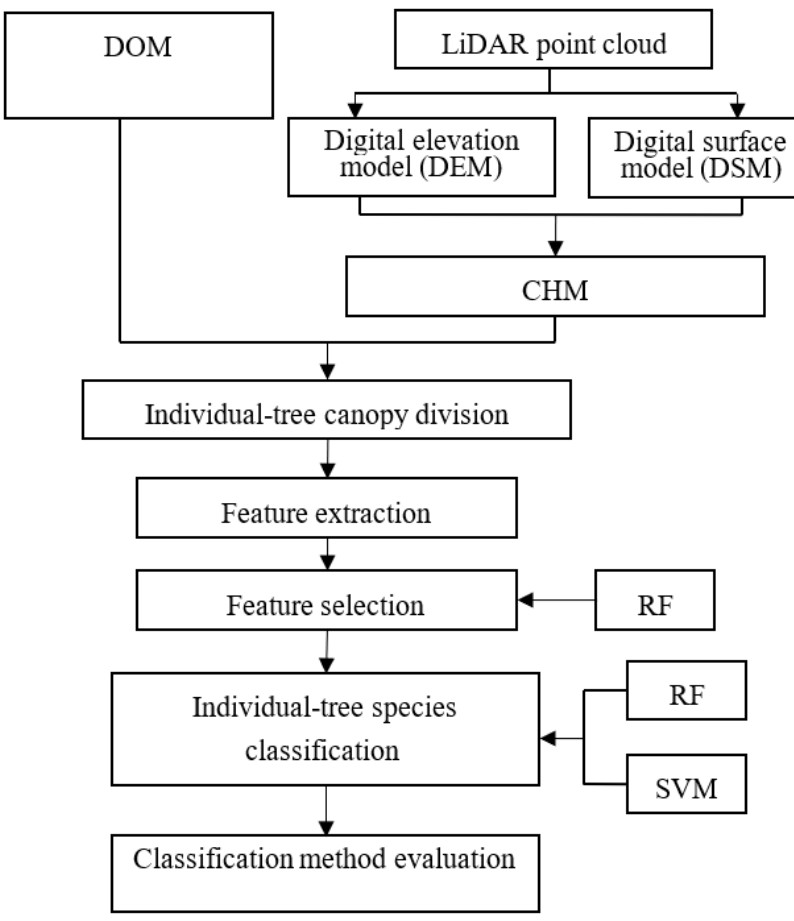

**Figure 3.** Technical process of tree species classification.

2.3.3. Feature Extraction

In this study, three types of features are extracted based on DOM: spectral, textural, and geometric features. Thereafter, point cloud and CHM features are extracted based on LIDAR point clouds. The detailed list is shown in Tables 2–6.

**Table 2.** Spectral features.

| Spectral Features | Feature Description | Symbolic Representation |
|---|---|---|
| Mean | Average pixel value of an object in a certain band | Rmean, Gmean, Bmean [1] |
| Standard deviation | Degree of dispersion of the gray value of pixels in the object area | Rstd, Gstd, Bstd [2] |

[1] Rmean, Gmean, and Bmean represent the mean values of the red, green, and blue bands, respectively. [2] Rstd, Gstd, and Bstd represent the standard deviation of each band of red, green, and blue, respectively.

**Table 3.** Texture features.

| Texture Features | Feature Description | Symbolic Representation |
|---|---|---|
| Homogeneity | Homogeneity of the image | Rhom3 (5,7,9,11), Ghom3 (5,7,9,11), Bhom3 (5,7,9,11) [3] |
| Contrast | Quality of image sharpness and depth of texture grooves | Rcon3 (5,7,9,11), Gcon3 (5,7,9,11), Bcon3 (5,7,9,11) [3] |
| Difference | Texture feature of the local image area | Rdis3 (5,7,9,11), Gdis3 (5,7,9,11), Bdis3 (5,7,9,11) [3] |
| Information entropy | Randomness measure of all information | Rent3 (5,7,9,11), Gent3 (5,7,9,11), Bent3 (5,7,9,11) [3] |
| Second order | Uniformity of gray distribution of image and thickness of texture | Rsec3 (5,7,9,11), Gsec3 (5,7,9,11), Bsec3 (5,7,9,11) [3] |
| Correlation | Similarity of image gray levels | Rcor3 (5,7,9,11), Gcor3 (5,7,9,11), Bcor3 (5,7,9,11) [3] |

[3] These symbolic represent the texture characteristics of each of the red, green, and blue bands at different window sizes.

**Table 4.** Geometric features.

| Geometric Features | Feature Description | Symbolic Representation |
|---|---|---|
| Area | Area of segmented object | Area |
| Perimeter | Perimeter of segmented object | Perimeter |
| Area perimeter ratio | Ratio of area of segmented object to perimeter | A_P |

**Table 5.** Point-cloud features.

| Point-Cloud Features | Feature Description | Symbolic Representation |
|---|---|---|
| Cumulative height percentile | Calculation of cumulative height percentile at 10% interval and calculation of its values at 25% and 75% intervals | H1, H10, H20, H25, H30, H40, H50, H60, H70, H75, H80, H90, H99 [4] |
| Height percentile | Calculation of height percentile at 10% intervals and calculation of its values at 25% and 75% intervals | HP1, HP10, HP20, HP25, HP30, HP40, HP50, HP60, HP70, HP75, HP80, HP90, HP99 [5] |
| Cumulative intensity percentile | Calculation of cumulative echo intensity percentile at 10% interval and calculation of its values at 25% and 75% intervals | INT1, INT10, Int20, Int25, Int30, Int40, Int50, Int60, Int70, Int75, Int80, Int90, Int99 [6] |
| Intensity percentile | Calculation of the percentile of echo intensity at 10% interval and calculation of its values at 25% and 75% intervals | IntP1, IntP10, IntP20, IntP25, IntP30, IntP40, IntP50, IntP60, IntP70, IntP75, IntP80, IntP90, IntP99 [7] |
| Mean intensity | Mean intensity of all echoes | INTmean |
| Intensity standard deviation | Intensity standard deviation of all echoes | INTstd |
| Intensity variance | Intensity variance of all echoes | INTvar |

[4] These symbols represent the cumulative height percentile at different heights. [5] These symbols represent the height percentile at different heights. [6] These symbols represent the cumulative intensity percentile at different heights. [7] These symbols represent the intensity percentile at different heights.

**Table 6.** CHM features.

| CHM Features | Feature Description | Symbolic Representation |
|---|---|---|
| Mean | Mean height of divided tree canopy | Hmean |
| Maximum | Maximum height of divided tree canopy | Hmax |
| Minimum | Minimum height of split canopy | Hmin |
| Standard deviation | Standard deviation of height of divided tree canopy | Hstd |
| Variance | Division of height variance of canopy | Hvar |
| Slope | Division of the slope of the canopy | Hslope |

2.3.4. Feature Selection Based on the RF Algorithm

A large number of features bring about the problem of redundancy. Even a classifier that is not sensitive to dimensionality decreases the classification accuracy, and feature screening can solve this problem [22]. This study selects the RF algorithm for feature screening because the RF algorithm can sort the importance of variables before classification [23]. The most important features to participate in the classification must be retained to solve the problem of excessive original features. The specific steps are the following:

First, the Gini index is calculated for each node $k$ in each tree:

$$G_k = 2\hat{p}_k(1 - \hat{p}_k) \tag{1}$$

$G_k$ represents the Gini index at node $k$. $\hat{p}_k$ represents the estimated value of the probability that the sample belongs to any class at node $k$.

The importance of a node is determined by the amount of change in the Gini index before and after the node is split:

$$I_{\Delta k} = G_k - G_{k1} - G_{k2} \tag{2}$$

$G_{k1}$ and $G_{k2}$ represent the child nodes generated by $G_k$. For each tree in the forest, the preceding criteria are used to recursively generate $I_{\Delta k}$.

Finally, samples and variables are randomly selected to generate a forest. It is assumed that the forest produces a total of T trees.

In the forest, if the variable $X_i$ appears $M$ times in the t-th tree, then the importance of the variable $X_i$ in the $t$-th tree is

$$I_{it} = \sum_{j=1}^{M} I_{\Delta j} \tag{3}$$

Then, the variable importance of $X_i$ in the entire forest is

$$I_{(i)} = \frac{1}{n} \sum_{t=1}^{T} I_{it} \tag{4}$$

Finally, the variables are selected according to the importance of the variables.

2.3.5. Tree Species Classification Based on Machine Learning

According to field survey data, the main tree species in the study area are paulownia, oak, fir, moso bamboo, maple poplar, and others. The final classification system is divided into four categories, namely, paulownia, oak, fir, and other tree species (including bamboo, maple poplar, shrubs, and other relatively small tree species).

The RF algorithm integrates a large number of trees into a forest, avoiding the one-sidedness and inaccuracy caused by the classification of a single decision tree, while the SVM does not require large samples and has great advantages in high-dimensional feature recognition. Therefore, this study applies RF and SVM in tree species classification.

The main steps of RF-based tree species classification are the following: (1) Random samples are created. Each time with replacement, n samples are drawn from the original sample set, and k extractions are performed in total. (2) A decision tree is established. In each process of generating a decision tree, from the D features in the feature space, d (d < D)

features are selected to form a new feature set, and the new feature set is used to generate a decision tree. (3) The generated k decision trees are combined, and the classification results of multiple decision trees are selected to obtain the final classification category.

The tree species classification process based on SVM is transforms the non-linear sample space into a linear space through the kernel function to realize the division of samples. In this study, the kernel function chooses the radial basis function [24], which is expressed as

$$k(x, x_i) = \exp\left(-\frac{\|x - x_i\|^2}{\delta^2}\right) \tag{5}$$

In the formula, $x$ and $x_i$ refer to the unknown vector and the support vector, respectively, and $\delta$ is the width of the function.

Based on the segmented image objects and the extracted features, 12 combinations are formed. These twelve combination schemes are shown in Table 7. When DOM is used, schemes I and II are unfeatured screening that use RF and SVM classifiers, respectively, whereas schemes III and IV are featured screening that use RF and SVM classifiers, respectively, after selection. When LiDAR is used, schemes V and VI are unfeatured screening that use RF and SVM classifiers, respectively, whereas schemes VII and VIII are featured screening that use RF and SVM classifiers, respectively, after selection. When LiDAR and DOM are used, schemes IX and X are unfeatured screening that use RF and SVM classifiers, respectively, whereas schemes XI and XII are featured screening that use RF and SVM classifiers, respectively, after selection.

**Table 7.** Classification scheme.

| Scheme | Feature Select | Type of Data | Classifier |
|--------|---------------|--------------|------------|
| I | No | DOM | RF |
| II | No | DOM | SVM |
| III | Yes | DOM | RF |
| IV | Yes | DOM | SVM |
| V | No | LiDAR | RF |
| VI | No | LiDAR | SVM |
| VII | Yes | LiDAR | RF |
| VIII | Yes | LiDAR | SVM |
| IX | No | DOM, LiDAR | RF |
| X | No | DOM, LiDAR | SVM |
| XI | Yes | DOM, LiDAR | RF |
| XII | Yes | DOM, LiDAR | SVM |

### 2.3.6. Accuracy Evaluation Indicators

In this study, stratified sampling is used to randomly select 40% of the data from each tree species for inspection. A total of 232 training samples and 155 test samples are available in the sample plots.

After obtaining the tree species classification results of different schemes, we need to verify the correctness to evaluate the effect of the individual-tree species classification of each scheme. The stratified sampling method is adopted, and the verification samples are selected through a combination of field investigation and visual interpretation. Constructing a confusion matrix is a common method to quantify classification accuracy [25]. In addition, MAE is selected for metrics in this study [26–28]. The indicators used to measure are shown in Table 8.

**Table 8.** Evaluation index of classification accuracy.

| Evaluation Index | Calculation Formula | Indicator Description |
|---|---|---|
| user accuracy, *UA* | $UA = \frac{x_{ii}}{x_{i+}}$ | Ratio of number of samples correctly classified into category i to the total number of samples in category *i* in the classification result, which reflects the reliability of a certain category being correctly identified |
| producer accuracy, *PA* | $PA = \frac{x_{ii}}{x_{+i}}$ | Ratio of the number of correct classifications of a category to the total number of that category in the reference sample |
| overall accuracy, *OA* | $OA = \frac{\sum\limits_{i=1}^{r} x_{ii}}{N}$ | Proportion of correctly classified samples to the total sample, reflecting the consistency between the classification results and the actual features |
| Kappa coefficient | $K = \frac{N\sum_{i=1}^{r} x_{ii} - \sum_{i=1}^{r}(x_{i+}x_{+i})}{N^2 - \sum_{i=1}^{r}(x_{i+}x_{+i})}$ | A precision statistic used to determine the matching degree between the actual feature category and classification result, which can weaken the influence of sample selection on the accuracy verification |
| *MAE* | $MAE = \frac{1}{N}\sum_{i=1}^{N}|y_i - \hat{y}_i|$ | Measure of the difference between the predicted and actual values of the model. |

$x_{ii}$ is the number of samples that were correctly classified. $x_{i+}$ is the total number of samples classified into class *i*. $x_{+i}$ is the total number of samples in class *i* in the reference samples. *r* is the total number of classes. *N* denotes the total number of samples drawn. $y_i$ is the actual expected output, and $\hat{y}_i$ is the model prediction.

## 3. Results

### 3.1. Optimized CHM Extraction Results

Due to the small canopy width, the use of a 3 × 3 filter window can retain the original information to the greatest extent. This study uses a 3 × 3 filter window to perform median smoothing filtering of the original CHM raster data. Comparing the local effect map of the median filter algorithm (Figure 4), we find many discontinuously distributed low values at the edge of the canopy in the original image. The image after median filtering is smoother, and invalid values in the image can also be removed effectively. Therefore, the median filter is selected to smooth the CHM data to reduce the impact of invalid values on accuracy. As shown in the final canopy height model in Figure 5, as the height of the canopy increases, and the image shows a brightness change from black to white. Figure 4b shows a partial demonstration of Figure 5.

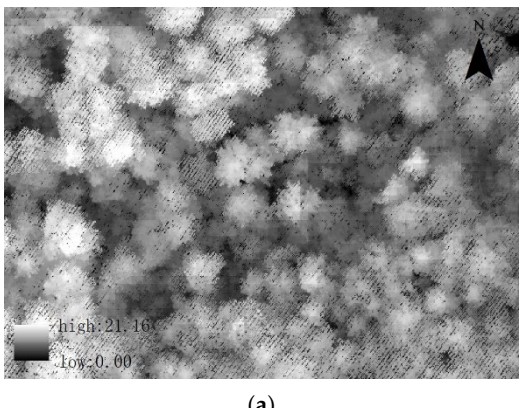

(**a**)

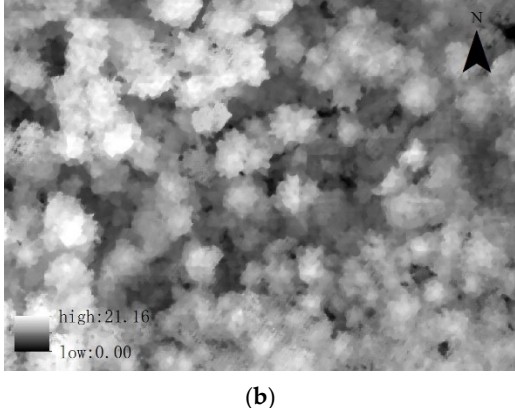

(**b**)

**Figure 4.** CHM in study area: (**a**) before optimization and (**b**) after optimization.

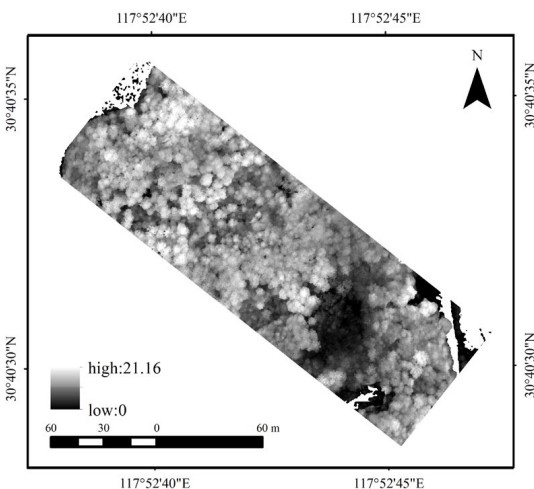

**Figure 5.** Optimized canopy height model of the study area.

### 3.2. Individual Tree Segmentation Results

The optimized CHM is segmented by the watershed segmentation algorithm. In combination with the field survey, the optimized results of partial tree crown segmentation and selected samples are shown in Figure 6.

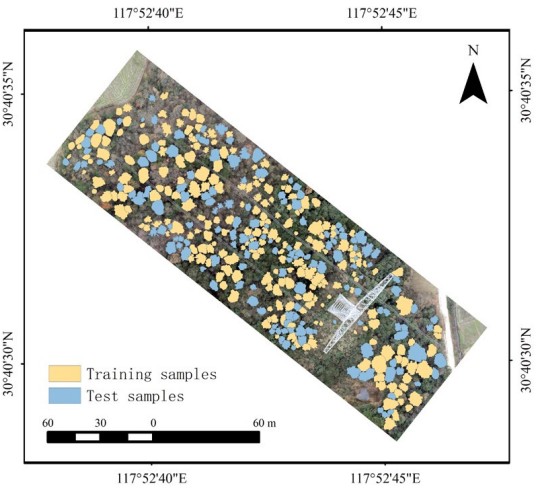

**Figure 6.** Sample crown in the study area.

### 3.3. Feature Screening Results

In this study, the RF algorithm is used to sort and filter the importance of a feature set composed of five types of 160 features based on DOM and LiDAR point-cloud data extraction. In total, 15 and 13 features were retained by RF screening when using only LiDAR and DOM, respectively, and 18 features were retained by feature screening after combining the two types of data. The ranking of the importance of the features retained after screening is shown in Figure 7. Analysis of feature importance revealed that the spectral mean and standard deviation scores for each band in the spectral features were the most stable and contributed the most, whether the classification was performed using only DOM or DOM combined with LiDAR. The texture features also have important contributions in the classification, where the contrast and correlation are the top ranked features in importance among the texture features. In the combination of DOM and LiDAR, point-cloud features, CHM features and geometric features all have more important roles in the classification.

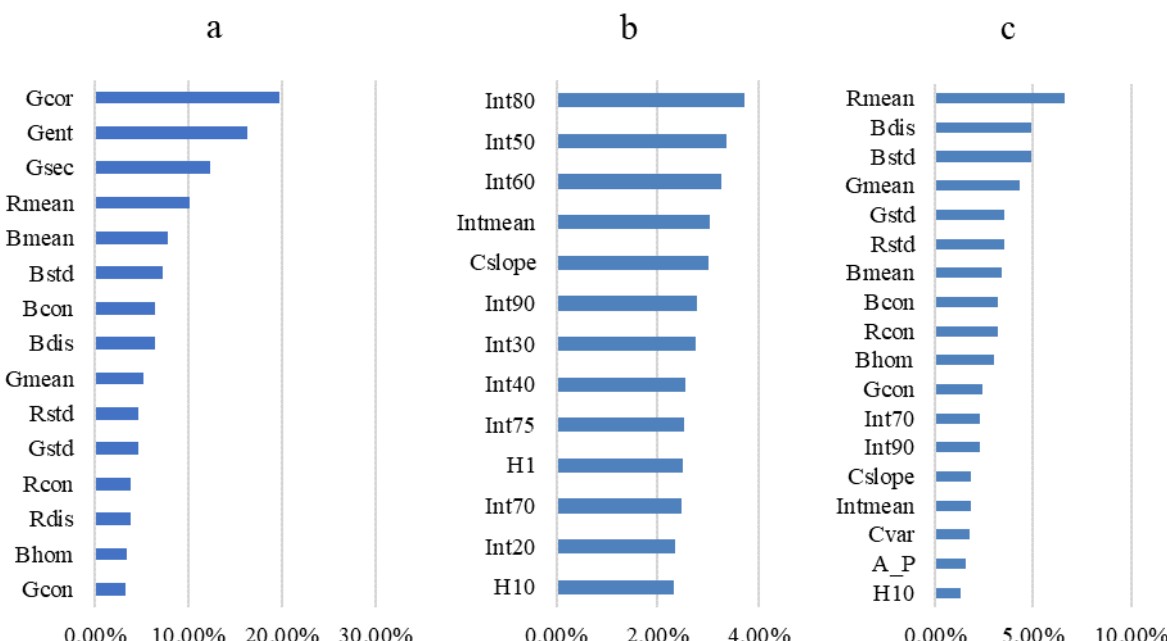

**Figure 7.** Results of feature selection by RF and importance ranking: (**a**) DOM extraction feature sorting results, (**b**) LiDAR extraction feature sorting results and (**c**) DOM and LiDAR extraction feature sorting results.

### 3.4. Classification Results and Accuracy Evaluation of Individual Tree Species

According to the results of the previous individual tree crown segmentation and feature extraction, the individual tree species are classified based on the designed four schemes, and the classification algorithm is implemented using Python. Samples are selected through a combination of field investigation and visual interpretation. Then, 60% of the data are selected as the training set for training the model, and 40% of the verification data are used to test the model reliability. After the tree species classification results are obtained, the test samples are selected to evaluate the accuracy of the results, and the best classification scheme is determined after analysis and comparison. The classification accuracy is shown in Table 9. The results of classifying trees according to the scheme 12 with the highest overall accuracy are shown in Figure 8.

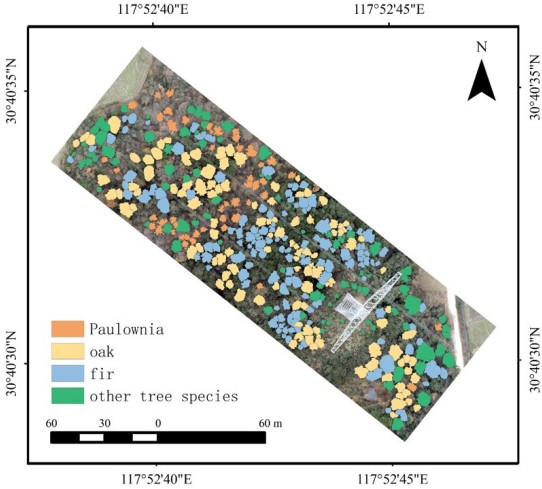

**Figure 8.** Classification results of tree species.

**Table 9.** Evaluation of classification accuracy.

| Scheme | Accuracy (%) | Paulownia | oak | fir | Other Tree Species | OA (%) | Kappa | MAE |
|---|---|---|---|---|---|---|---|---|
| I | PA | 70.00 | 87.50 | 66.67 | 69.44 | 74.19 | 0.66 | 0.39 |
| | UA | 77.78 | 76.36 | 72.34 | 71.43 | | | |
| II | PA | 60.00 | 83.33 | 64.71 | 72.22 | 71.61 | 0.60 | 0.42 |
| | UA | 75.00 | 67.80 | 73.33 | 74.29 | | | |
| III | PA | 80.00 | 89.58 | 64.71 | 77.78 | 79.35 | 0.71 | 0.29 |
| | UA | 88.89 | 74.14 | 78.26 | 84.85 | | | |
| IV | PA | 55.00 | 85.42 | 70.59 | 75.00 | 73.54 | 0.63 | 0.41 |
| | UA | 84.62 | 77.36 | 71.43 | 67.50 | | | |
| V | PA | 55.00 | 70.83 | 45.10 | 47.22 | 52.26 | 0.34 | 0.74 |
| | UA | 55.00 | 57.63 | 52.27 | 53.13 | | | |
| VI | PA | 55.00 | 56.25 | 54.90 | 36.11 | 50.97 | 0.39 | 0.73 |
| | UA | 52.38 | 55.10 | 44.44 | 59.10 | | | |
| VII | PA | 45.00 | 66.67 | 43.14 | 55.56 | 53.55 | 0.36 | 0.69 |
| | UA | 50.00 | 51.61 | 52.38 | 60.61 | | | |
| VIII | PA | 35.00 | 58.70 | 45.10 | 77.78 | 54.84 | 0.39 | 0.71 |
| | UA | 58.33 | 58.70 | 56.10 | 56.00 | | | |
| IX | PA | 80.00 | 83.33 | 76.47 | 80.56 | 80.00 | 0.73 | 0.36 |
| | UA | 72.73 | 78.43 | 73.58 | 100.00 | | | |
| X | PA | 75.00 | 81.25 | 84.31 | 69.44 | 78.21 | 0.70 | 0.30 |
| | UA | 83.33 | 82.98 | 72.88 | 80.65 | | | |
| XI | PA | 90.00 | 85.75 | 74.51 | 86.11 | 83.23 | 0.77 | 0.23 |
| | UA | 85.71 | 77.78 | 82.61 | 91.18 | | | |
| XII | PA | 85.00 | 85.42 | 86.27 | 83.33 | 85.16 | 0.79 | 0.21 |
| | UA | 89.47 | 89.13 | 80.00 | 85.71 | | | |

### 3.5. Results Analysis

Analysis of the accuracy of the scenarios based on the data in Table 9 shows that:

(1) When using DOM only, scheme III had the highest classification accuracy with an overall accuracy of 79,35%, Kappa coefficient of 0.71, and MAE of 0.29. After feature selection, the accuracy of both classifiers improved. The classification schemes with feature selection improved the accuracy of classification using RF and SVM by 5.16% and 1.93%, respectively, compared to the schemes without feature selection.

(2) When using LiDAR only, none of the classification results of schemes V–VIII were very good, and none of the overall accuracies reached 55%. For this study area, the effect of using LiDAR only for tree species classification was not satisfactory.

(3) When using the combination of DOM and LiDAR for classification, scheme 12 had the best classification results, with an overall accuracy of 85.16% and a Kappa coefficient of 0.79. The accuracy of classification using RF and SVM improved by 3.23% and 6.45%, respectively, after feature selection compared to that in the scheme without feature selection.

(4) In terms of tree species, *Paulownia* was more affected by feature selection, and in most cases, PA, UA improved after feature selection. Oak and fir were more affected by feature selection when LiDAR and DOM were combined for classification, and there was a significant improvement in PA and UA. The classification accuracy of other tree species was not ideal due to more internal species, and it may be necessary to classify other tree species into several more detailed categories in order to improve the accuracy.

## 4. Discussion

### 4.1. The Impact of Feature Screening on Classification

Feature screening is very important for classification research. Feature screening can reduce multicollinearity among features and improve computational efficiency and classification accuracy. The results show that the accuracy and Kappa coefficient of RF and SVM classification improved after feature screening, and RF feature screening achieved good results in both RF and SVM classification. Therefore, the RF signature screening is reliable. Using multispectral and LiDAR data for classification, Pham et al. [29] explored the role of RF signature screening for classification. When the multi-source data were combined, the AO after RF screening reached 85.4%, and the Kappa coefficient was 0.81, which were 0.05 and 0.07 higher than those without feature screening, which is very similar to the results of this study.

### 4.2. The Impact of the Classification Algorithm on the Accuracy

For this study, when DOM was combined with LiDAR for classification, the SVM algorithm was more accurate after feature filtering. This may be because the SVM model can solve high-dimensional problems well and is better for machine learning in the case of small samples. The RF algorithm has been shown to overfit in some noisy classification or regression problems.

### 4.3. Contribution of Different Features to Classification

When DOM was combined with LiDAR for classification, intensity and height features were extracted from LiDAR, spectral and texture features are extracted from DOM, and the performance of these features was evaluated. The results show that the spectral features contributed the most to the classification. Among them, the green band was very important in distinguishing tree species, probably because of the different pigment contents of different tree species; the contents of chlorophyll, carotenoid, anthocyanin, and lutein are closely related to the reflectance of the green band. Texture features also contributed greatly, such as the contrast and correlation within the convolution kernel. Texture features are global features that can describe the surface properties of the scene corresponding to the image area, so they have great potential for classification. The LiDAR point cloud features provided three-dimensional information of trees for classification. The first echo intensity features and height features of LiDAR data were sensitive to canopy conditions, well represented the tree canopy structure and morphological features, and contributed greatly to tree species classification.

### 4.4. Effect of Observation Season on the Classification Accuracy

Huaipeng Liu [30] classified urban tree species based on four seasons of RedEdge-MX data, and the results showed that among the four seasons of the year, the classification of tree species based on spring data was the best. The accuracy of tree species classification can be improved by combining data from two, three, and four seasons. Other studies on tree species classification were conducted in summer or autumn and also achieved good accuracy, very similar to the results of the present study [31,32]. In future studies, more data from different periods will be applied to the study of tree species classification so that the relationship between seasons and the accuracy of tree species classification can be discussed in more depth.

## 5. Conclusions

To solve the problem of tree species classification in transmission line corridors, this study used multi-source UAV data and ML methods to effectively overcome the problem of low tree species classification accuracy and realized the extraction and classification of individual trees in transmission line corridors. The results show that feature selection is an important task in classification research on tree species. After feature screening, the accuracy and kappa coefficient of RF and SVM classification improved. Thus, RF feature

screening achieved good results in both RF and SVM classification, which shows that this type of feature screening is reliable.

During the experiment, the extraction of features was the most important, and the contribution of various features to the classification results was different. The research results show that spectral features contributed the most to classification. In addition, texture features played a very important role in classification, such as the correlation and contrast in the convolution kernel of the green band and blue band. The features extracted from LiDAR data were used to supplement the 3D information of the individual tree and were also indispensable in the classification. The research results show that the first echo intensity feature and height feature of LiDAR data also had a high contribution to the classification. In future research, more data sources will be selected to achieve large combinations so that more effective features can be extracted to distinguish tree species. This will provide important information for the establishment of an intelligent early warning system for tree barriers in transmission line corridor areas, thus enabling sustainable management of forest resources and effective monitoring of species diversity in these corridors.

**Supplementary Materials:** The following supporting information can be downloaded at: https://www.mdpi.com/article/10.3390/su14148273/s1, File S1: Data.

**Author Contributions:** Conceptualization, X.L. and R.W.; methodology, X.L.; software, X.L.; validation, X.L., R.W. and X.C.; formal analysis, X.L.; investigation, X.L., X.C., Y.L. and Y.D.; resources, R.W.; data curation, Y.L. and Y.D.; writing-original draft preparation, X.L.; writing-review and editing, X.L.; visualization, X.C. and Y.L.; supervision, R.W.; project administration, R.W.; funding acquisition, R.W. All authors have read and agreed to the published version of the manuscript.

**Funding:** This research was supported by the National Natural Science Foundation of China: 'biomass precision estimation model research for large-scale region based on multi-view heterogeneous stereographic image pair of forest' (Grant No. 41971376).

**Institutional Review Board Statement:** Not applicable.

**Informed Consent Statement:** Not applicable.

**Data Availability Statement:** Not applicable.

**Acknowledgments:** We acknowledge the financial support from the National Natural Science Foundation of China: 'biomass precision estimation model research for large-scale region based on multi-view heterogeneous stereographic image pair of forest' (Grant No. 41971376). We are sincerely grateful for the efforts of Ruirui Wang, Xingwang Chen, Yiran Li, Yunshan Duan, and other colleagues for their help in field and laboratory studies.

**Conflicts of Interest:** The authors declare no conflict of interest.

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
