# Peer review of "Classification of Transmission Line Corridor Tree Species Based on Drone Data and Machine Learning"

_sustainability, doi:10.3390/su14148273_

Round 1

Reviewer 1 Report

The authors present a study where a combination of airborne remote sensing source are used to classify trees in the surroundings of electricity transmission lines. The article may be of interest for readers concerned with the use of remote sensing for tree monitoring and tree classification. Some comments/suggestions are listed hereafter:

Line 28: The introduction starts with "The introduction should briefly..." It seems to be an error.

Line 103: Several acronyms are used in the text. Please indicate the meaning of acronyms (DOM, CHM) the first time the terms appear in the text.

Figures 4 and 5: Is it possible to add a colorbar to Figures 4 and 5 in order to understand the actual meaning of the gray tones.

Line 128: Vincent: Provide reference in the text.

The study is based on observations conducted in June 2016. Since the document proposes a methodological approach for a particular application, it would be interesting to read a discussion on the importance (if any) of the timing of the observations with respect to the annual growth patterns of vegetations. Would the results be different if the observations were conducted in a different season?

Perhaps the most important remark: The title of the article suggests that the content would deal with situations that are specific to transmission lines. However, the methods and results seem to be of general application for any situation in which one needs to identify tree species in a forest. No explanation is given on the relevance of identifying tree species for a proper maintenance of transmission lines, for instance. It can be assumed that it suffices to monitor the vegetative growth of trees (height) and/or monitor the spread of trees to tracks that should be free of trees. Why is it important to know the tree species for the purpose of maintaining transmission lines? If this point can not be clarified, consider changing the title of the article such that it better matches the content.

Reviewer 2 Report

  1. The abstract needs to add numerical results for the accuracy of the classifier and also for evaluation measures.
  2. A contribution of research needs more clarification in the introduction section.
  3. Add a separate section for discussion and comparison of results.
  4. Use more evaluation measures for evaluation of classification accuracy
  5. References are updated for the years 2021 and 2022, and you can use

https://doi.org/10.1016/j.desal.2021.115411

https://doi.org/10.1016/j.compbiomed.2021.104606

https://doi.org/10.1016/j.aej.2022.03.050

Round 2

Reviewer 1 Report

Dear authors,

Thank you for the modifications made to the original manuscript.

Reviewer 2 Report

Accept in present form

Reviewer 3 Report

The revised manuscript responds to all my concerns, and I recommend it to be accepted and published.